# Femtosecond Laser Modification of Silica Optical Waveguides for Potential Bragg Gratings Sensing

**DOI:** 10.3390/ma15186220

**Published:** 2022-09-07

**Authors:** Jian Chen, Ji-Jun Feng, Hai-Peng Liu, Wen-Bin Chen, Jia-Hao Guo, Yang Liao, Jie Shen, Xue-Feng Li, Hui-Liang Huang, Da-Wei Zhang

**Affiliations:** 1Shanghai Key Laboratory of Modern Optical System, Engineering Research Center of Optical Instrument and System, Ministry of Education, School of Optical-Electrical and Computer Engineering, University of Shanghai for Science and Technology, Shanghai 200093, China; 2Key Laboratory of High Field Laser Physics and CAS Center for Excellence in Ultra-Intense Laser Science, Shanghai Institute of Optics and Fine Mechanics, Chinese Academy of Sciences, Shanghai 201800, China; 3Shanghai Honghui Optics Communication Tech. Corp., Shanghai 201822, China

**Keywords:** gratings, ultrafast, integrated optics devices, waveguides, microstructure fabrication

## Abstract

The optimum femtosecond laser direct writing of Bragg gratings on silica optical waveguides has been investigated. The silica waveguide has a 6.5 × 6.5 µm^2^ cross-sectional profile with a 20-µm-thick silicon dioxide cladding layer. Compared with conventional grating inscribed on fiber platforms, the silica planar waveguide circuit can realize a stable performance as well as a high-efficiency coupling with the fiber. A thin waveguide cladding layer also facilitates laser focusing with an improved spherical aberration. Different from the circular fiber core matching with the Gaussian beam profile, a 1030-nm, 400-fs, and 190-nJ laser is optimized to focus on the top surface of the square silica waveguide, and the 3rd-order Bragg gratings are inscribed successfully. A 1.5-mm long uniform Bragg gratings structure with a reflectivity of 90% at a 1548.36-nm wavelength can be obtained. Cascaded Bragg gratings with different periods are also inscribed in the planar waveguide. Different reflection wavelengths can be realized, which shows great potential for wavelength multiplexing-related applications such as optical communications or sensing.

## 1. Introduction

Wavelength-division multiplexing (WDM) devices can effectively improve data transmission bandwidth and have been extensively utilized in optics communications and multiparameter sensings [1,2,3]. Several approaches for WDM have been investigated, including Mach–Zehnder interferometers (MZIs) [4,5], microring resonators [6], arrayed waveguide gratings (AWGs) [7,8], thin film filters (TFFs) [9], and Bragg gratings [10]. MZIs and microring-based devices are usually limited by free spectral range (FSR) and channel bandwidth. For tunable AWGs, different approaches have been proposed in order to relax the limitations imposed by wavelength tolerances [11]. Multimode waveguides, double-peaked electric field distribution, and thermo-optic effect can be used for wavelength tuning, but they will increase the insertion loss and fabrication complexity [8]. TFFs have the advantages of a narrow passband, low loss, and good temperature stability. However, the fabrication process is a little complex and sometimes more than 100 layers may need to be coated [12,13]. The Bragg grating can reflect the designed wavelength and is considered a promising candidate for WDM systems. With the development of integrated photonic circuit technology, silicon waveguide-based Bragg grating has received more attention. However, its high coupling and polarization-dependent loss as well as the temperature sensitivity are not so favorable [14]. For the silica waveguide, the cross-sectional profile is usually square, which can work polarization independently. The waveguide mode matches the pattern of single-mode fiber, resulting in a low coupling loss [15]. The silica waveguide-based AWGs have been developed by the typical semiconductor lithographic process, which requires a high precision alignment. Furthermore, it is not so convenient to adjust the grating parameter for manufacturing error compensation. On the other hand, femtosecond laser direct writing technology can realize three-dimensional, low-cost, and flexible waveguide fabrication, which can induce permanent refractive index changes in transparent materials [16,17].

Till now, a large number of femtosecond laser-inscribed Bragg gratings have been reported in fibers [18,19]. The stacking inscription technique has been adopted to prepare the highest reflectivity Bragg grating for mid-infrared applications [20]. However, the cylindrical geometry of the fiber would cause an aberration when the laser is focused inside with air-based lenses, and adaptive optics aberration compensation should be adopted [21]. The aberrations can also be improved by placing fibers in refractive index matching fluids or ferrules, which are not so convenient and are unsuitable for mass manufacturing [22,23]. Some Bragg gratings are also realized by irradiating the fiber core using the laser interference pattern generated by the phase mask without eliminating spherical aberration. However, a phase mask has a fixed structure parameter and can only work for a specific resonance wavelength, which is slightly expensive and time-consuming [24]. Nevertheless, maintaining the roundness, symmetry, and mode–field profile of the fiber grating by the femtosecond laser direct writing is still challenging. A silica-based planar lightwave circuit (PLC) can naturally avoid aberration during laser writing, which has stable performance and can be mass-produced [25,26]. However, the femtosecond laser direct writing of the silica-based PLC waveguide for Bragg gratings has not yet been reported, to the best of our knowledge.

In the following, the optimum preparation of Bragg gratings on silica waveguides by femtosecond laser direct writing is presented. Compared with the fiber Bragg grating, the waveguide case can be easily prepared with no need for aberration correction. The fabrication reproducibility is competitive and the device performance is also stable. Most importantly, the waveguide gratings can be monolithically integrated with other photonic integrated devices, which will benefit some system-on-chip applications. The 3rd-order Bragg gratings with a length of 1.5 mm are fabricated with an extinction ratio of 9.5 dB and reflectivity of about 90% at a 1548.36-nm resonance wavelength. Moreover, cascaded Bragg gratings are also inscribed to achieve multi-wavelength reflection. Detailed experimental results and interaction mechanisms are analyzed and discussed.

## 2. Bragg Grating Design and Fabrication

The silica chip was prepared by photolithography and reactive ion etching on a 4-inch wafer. A 20-µm-thick SiO_2_ cladding layer was then deposited by flame hydrolysis deposition (FHD), which could provide a convenient refractive index modification range and smooth film profile, a low propagation loss (0.01 dB/cm), low coupling and reflection loss (less than 0.1 dB) due to almost the same refractive index and mode field diameter as conventional single-mode fiber, excellent physical and chemical stability, as well as an inexpensive large-scale fabrication [27]. Then, the waveguides were cut to a size of 30 × 2.5 × 1 mm^3^, with facets polished and packaged with input/output fiber as shown in Figure 1a. The waveguide has a germanium-doped silica core with a 6.5-µm-wide square profile, as shown in Figure 1b. The refractive indices of the core and cladding layer are 1.463 and 1.444, respectively, at a wavelength of 1550 nm. Such a refractive index also benefits from femtosecond laser inscription [28]. The waveguide can work on polarization independently from the calculated effective refractive index of about 1.447 for both transverse electric (TE) and transverse magnetic (TM) modes, whose profiles can match well with that of the fiber, resulting in a low mode overlap loss and Fresnel reflection between chip and fiber. Tight packaging can also improve the Fresnel reflection. In the experiment, the actual connection loss of the silica-based packaged device was measured to be 0.2 dB, which includes coupling loss as well as propagation loss. Some commercial PLC devices can realize a coupling loss of less than 0.1 dB, whereas the current waveguide can be further optimized to realize a better fiber-to-chip coupling.

For the waveguide Bragg grating, it is a structure that periodically modulates the effective refractive index of a waveguide. When a wave enters the grating region, changes in refractive index can cause periodic reflections. If all the reflections are in phase, many reflections combine with constructive interference and a strong reflected signal will appear. That is Bragg resonance wavelength *λ_B_*, which can be expressed as [29]
(1)λB=2neffΛm,
where *n_eff_* is the effective refractive index of the propagating mode, Λ is the grating period, and *m* = 1, 2, 3, …, is the grating order. For uniform Bragg gratings, the maximum reflectivity *R*_max_ at the Bragg wavelength can be calculated by the transmission spectrum [23]
(2)$$R_{\max}=1-10^{-(T/10)},$$
where *T* is the transmittance in decibels (dB) at *λ_B_*. The coupling coefficient *κ* depends on the refractive index modulation (Δ*n*), mode overlap factor (*η*), and Bragg wavelength, which can be expressed as [30]
(3)κ=πΔnηλB.

The spectrum bandwidth of Bragg grating with length *L* can be obtained by [31]
(4)Δλ=λB2neffκ2π2+1L2.

The bandwidth of weakly modulated Bragg gratings (*κL* < 1) is length-limited, and a longer grating has a narrower bandwidth. In contrast, for strongly modulated Bragg gratings (*κL* > 3), the light does not penetrate the full length of the grating. Thus, the bandwidth is directly proportional to the coupling coefficient and almost independent of length. Therefore, by reducing the coupling coefficient, it is possible to narrow the linewidth of a strong modulation Bragg grating.

A schematic illustration of the femtosecond laser inscription process is shown in Figure 2a. The system consists of a 1030-nm femtosecond laser (YL-20, Anyang, China) with a pulse width of 400 fs and a repetition rate of 25–5000 kHz. After the collimated beam passes through the electric shutter, it is introduced to the upper gantry bracket by a mirror, then goes through a half-waver-plate, a dichroic mirror, and an objective lens (Mitutoyo (Kawasaki, Japan), 20×, NA 0.40). The lens has a working distance of 20.35 mm, which can focus the laser to a spot size of about 1 µm. The electric shutter is used to control the opening and blocking of the pulse train of the laser pulse. The dichroic mirror can transmit visible light and reflect light at a 1030-nm wavelength. A CCD camera with a coaxial light source is installed for real-time monitoring of the inscription process. Due to the transparency of the PLC chip, a transmitted illumination system is adopted. A three-dimensional air flotation platform (Aerotech Inc., Pittsburgh, PA, USA) is used for the motion control, where the objective lens is mounted on the Z-axis platform to facilitate the adjustment of the laser focal length, and the XY-axis air flotation platform is used for the motion of the PLC chip. Although the resolution of the air flotation platform can provide a fabrication resolution of 0.5 nm, the preparation resolution is about ±3 nm due to the influence of repeatability. Figure 2b shows the image of the PLC chip placed on the moving stage, and the waveguide can be displayed on the monitoring screen. The laser inscription system is integrated on a custom granite base, which is highly resistant to disturbance with excellent positioning stability.

After precisely adjusting the position of the silica chip in vertical and horizontal directions, the waveguides were inscribed with different laser energy, engraving speed, grating length, and focusing position. The transmission and reflection spectra of Bragg gratings were monitored in real-time during the inscription process by using an amplified spontaneous emission (ASE) laser, a circulator, and an optical spectrum analyzer (OSA) (Yokogawa (Tokyo, Japan), AQ6370C). It should be noted that when characterizing the polarization characteristic, the ASE light source needs to connect a polarizer and a polarization controller. TE light is presented for the performance characterization, though the device is polarization-independent. According to Equation (1), the grating periods should be 0.536, 1.071, and 1.607 µm for the 1st, 2nd, and 3rd grating orders, respectively, with consideration of the effective refractive index of 1.447 at a Bragg wavelength of 1550 nm. Though the coupling coefficient is higher for the low grating order [31], the short grating period usually needs an oil-immersion objective lens with a high numerical aperture such as 1.25, which greatly increases the system complexity. On the other hand, the grating can be inscribed more conveniently for the 3rd-order grating, whose coupling coefficient is also comparable to the lower-order gratings when the duty cycle reaches more than 75% [31]. The grating duty cycle is defined as the ratio of the width of the laser writing line to the grating period. For the inscribing of the waveguide grating, the laser energy, scanning speed, and focusing position should be optimized. The optimum 3rd-order Bragg gratings could be obtained with high reflectance and low loss at a laser power of 190 nJ, engraving speed of 0.15 mm/s, and focusing position just above the waveguide. Since the refractive index change is not so easy to be characterized by the scanning microscope, we measure the top-surface grating profile by a microscope as in Figure 2b. The silica-based PLC was placed in a marked position on the moving stage. There is a cut mark at the edge of the waveguide, which is used to ensure that the adjustment is started from the same position every time since we can clearly observe the mark through the imaging system. The pitch, rotation, and focusing positions were constantly adjusted so that the waveguide remained clearly imaged over the whole processing range. Then the platform moved to the initial mark position for laser writing. Good repeatability can be guaranteed and we can obtain almost the same performance when we repeated the process 10 times with the same laser writing parameters. It can also be confirmed by the consistency of the resonance wavelengths for the single and multi-grating structures.

## 3. Characterization of Uniform and Cascaded Bragg Gratings

The 3rd-order Bragg gratings were fabricated successfully as in the inset of Figure 2b, with a grating duty cycle of about 75% and a writing line length of 30 µm. When preparing a 3rd-order Bragg grating, the coupling coefficient increases with the duty cycle (50–75%), and the highest coefficient can be achieved with a duty cycle of 75% [31]. For the duty cycle, it should be optimized to obtain a maximum extinction ratio for the resonance peak. Here, it was adjusted with varying laser energy. A maximum extinction ratio of about 9.5 dB can be obtained with a laser energy of 190 nJ, corresponding to a duty cycle of about 75% [31], roughly matching the shape in the microscope image as in Figure 2b. The coupling efficiency affects the extinction ratio and reflectivity of the Bragg grating. The reflectivity of the Bragg resonance can be inferred from the observed transmission dip according to Equation (2). When the coupling coefficient is optimal, the extinction ratio and reflectance will increase with the length. During the grating writing, the optimal reflectivity can be obtained by observing the extinction ratio with the change of grating length. With the increase in the total grating length, the reflectivity and extinction ratio of the transmission peak will increase and then saturate at a certain length. More gratings will not improve the reflectance much at the Bragg wavelength but cause higher transmission loss. During the fabrication, the total length of the grating is optimized to be around 1.5 mm.

The transmission loss is slightly high here. Actually, the loss may be mainly caused by the laser fabrication condition such as the pulse width, laser energy, focusing position, etc., which may influence the grating profile and the relative position to the waveguide. The laser with too large energy will cause large waveguide loss or even damage the waveguide, whereas a too small energy laser is difficult to form refractive index modulation. The focusing position will influence the loss and the pulse width will affect the interaction mechanism. When the focusing depth is 20 to 25 μm, the laser energy has a wider adjustment range and is easy to form refractive index modulation [32]. We further optimized the focusing position and engraving speed to improve the bandwidth and extinction ratio of the Bragg gratings. The optimum 3rd-order Bragg gratings could be obtained with high reflectance and low loss at a focusing position of 20 µm and engraving speed of 0.15 mm/s, as shown in Figure 3. Further optimization is still needed and rapid thermal annealing may also help to improve the transmission loss [33]. The reflectance can reach 90% for the total grating length of 1.5 mm at a Bragg wavelength of 1548.36 nm. For the fabricated Bragg grating, the polarization-dependent loss is also very low and the device performs almost the same for both polarizations. A slightly different resonance wavelength from 1550 nm means that the refractive index change is about 1.449, which is also similar to the reported value [32].

Since good repeatability can be obtained, cascaded Bragg gratings can also be prepared. Actually, multi-wavelength multiplexing also has important applications in multiparameter sensing and optical communication [34]. The multiplexing of fiber Bragg gratings can be used for the multi-wavelength ring laser [35]. To demonstrate the potential of multiparameter sensing, a series of cascaded Bragg waveguide gratings were inscribed on a silica waveguide by the femtosecond laser direct writing technology. The cascaded grating structure consists of many uniform Bragg gratings with total length *L* and is separated by equal intervals *d*, as shown in Figure 4. Here, both the grating length and interval between two Bragg gratings are about 1.5 mm. Different gratings can be cascaded such as two gratings with periods of 1.60 and 1.61 μm, whose transmission spectra are shown in Figure 5a. The corresponding resonance wavelengths are 1543.8 and 1553.7 nm, respectively, with an insertion loss of about 2 dB. More gratings can also be prepared, such as 5 gratings with periods of 1.60, 1.61, 1.62, 1.63, and 1.64 μm. The obtained transmission spectrum is shown in Figure 5b. The corresponding resonance wavelengths are 1543.8, 1553.7, 1563.4, 1572.3, and 1581.4 nm, respectively, with a free spectral range of about 9.7 nm. The insertion loss increases to about 5 dB. The resonance wavelength of the grating with the same period is identical for the two and five cascaded cases, which confirms the good repeatability of the system. For cascaded gratings, the flatness of the chip needs to be carefully adjusted before the grating writing.

Multiphoton nonlinear absorption effect plays an important role in the femtosecond laser direct writing on a silica-based PLC waveguide by FHD processing [32]. Besides, there is still much space for the improvement of grating performance such as the loss by optimizing the femtosecond laser writing process. The successful preparation of the Bragg grating on the silica waveguide shows the potential of the post-trimming or even three-dimensional structure fabrication on a conventional two-dimensional planar waveguide. Compared with the fiber Bragg grating, the Bragg gratings on photonic integrated circuits are more stable and attractive [36]. Based on the waveguide structure, the femtosecond laser processing can easily achieve multi-wavelength multiplexing [37] or multiparameter sensing-related [38] applications. Bragg gratings are widely used as a sensor to measure refractive index, temperature, strain, and other parameters; any bending and strain of the grating section must be avoided when measuring refractive index and temperature, which is usually achieved by being placed in a capillary or straightened for a fiber case [39,40]. The presented PLC Bragg grating can avoid the cross-sensitivity caused by stress and bending, thus greatly simplifying the measurement and improving the detection sensitivity. For example, Zhang et al. proposed a highly sensitive waveguide magnetic field sensor device via laser direct writing, which is an order of magnitude higher sensitivity than similar sensors [41]. Cascaded grating structures with different periods on a planar waveguide platform can also achieve different information encoding, such as gray code [42,43]. Microchannels can be prepared on the top surface of each grating, which can be used for multichannel biosensors or multiparameter sensing, as illustrated in Figure 6. The sensor chip is connected to an ASE laser and OSA for characterization, which can be used to determine microbial function, sample concentration, detection, and cell screening [44,45]. However, for the real fabrication, the processing condition such as for optofluidic channel needs more time to optimize. Further work still needs to be conducted for practical applications. Nonetheless, the presented waveguide platform fabricated by semiconductor process and laser inscription can be applied in the fields such as lab-on-chip sensing [46], optical computing [47], and so forth.

## 4. Conclusions

To summarize, Bragg gratings were inscribed at the desired resonance wavelength in the silica PLC chips by femtosecond laser direct writing. A reflectivity of approximately 90% can be realized for the 3rd-order Bragg gratings at a wavelength of 1548.36 nm. Good repeatability can be obtained and the presented fabrication process is more convenient and suitable for mass production. In addition, cascaded Bragg gratings were also prepared on the silica waveguide with slight variations in the period. Multi-wavelength reflection can be obtained, which is beneficial for multiplexing or multiparameter sensing. The presented femtosecond laser post-trimming method will facilitate more complex structure fabrication for conventional semiconductor planar waveguide platforms.

## Figures and Tables

**Figure 1 materials-15-06220-f001:**
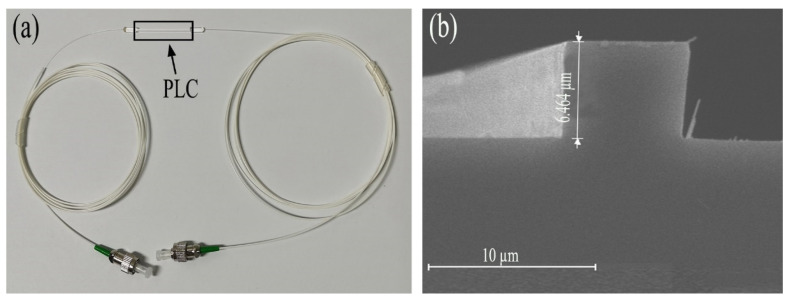
(**a**) Photo of fiber packaged silica chip. (**b**) Cross-sectional SEM image of the silica waveguide.

**Figure 2 materials-15-06220-f002:**
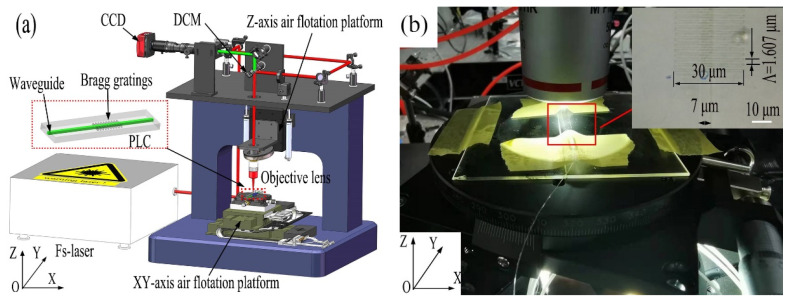
(**a**) Schematic illustration of Bragg gratings fabrication. (**b**) Photo for the PLC chip on the moving stage. Inset: the microscope image of the fabricated gratings.

**Figure 3 materials-15-06220-f003:**
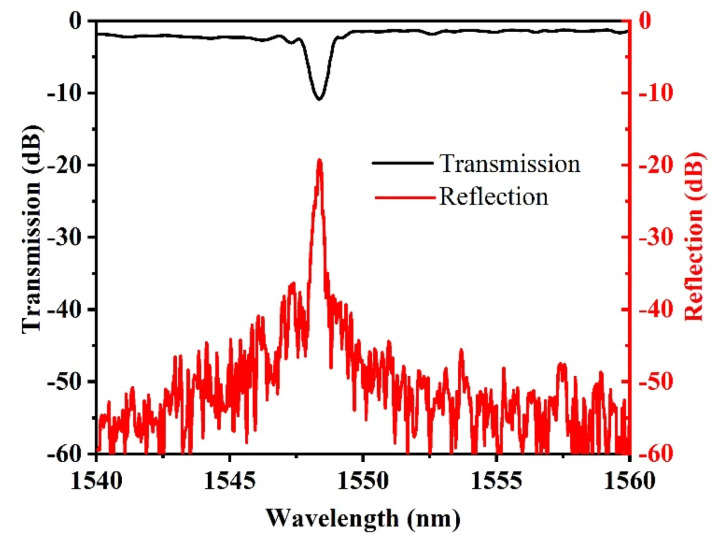
Measured transmission and reflection spectra of the Bragg grating.

**Figure 4 materials-15-06220-f004:**
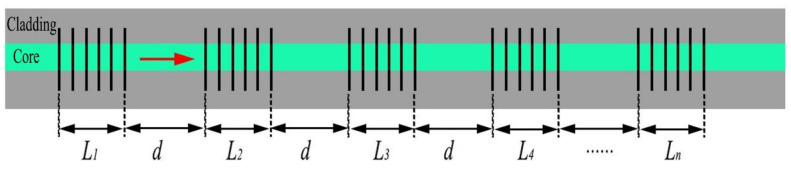
Schematic diagram of the cascaded Bragg gratings.

**Figure 5 materials-15-06220-f005:**
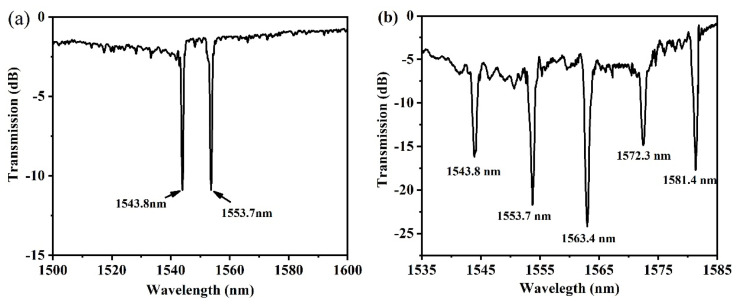
Transmission spectra of (**a**) two and (**b**) five cascaded Bragg gratings, respectively.

**Figure 6 materials-15-06220-f006:**
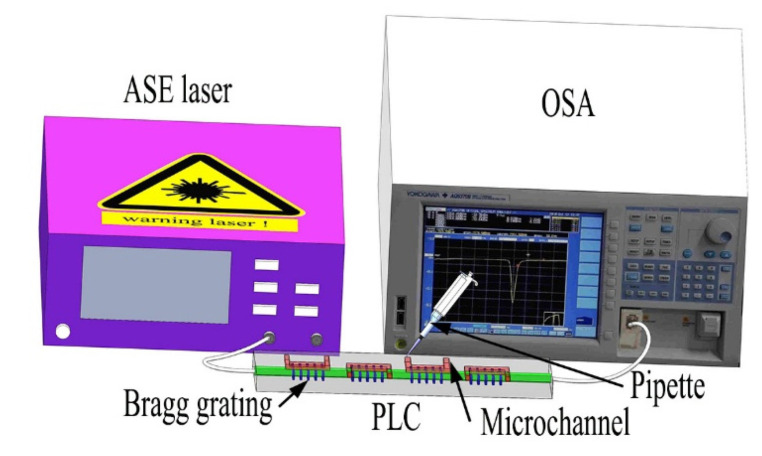
Schematic illustration of multiparameter sensing.

## Data Availability

Data supporting the results reported in this paper may be obtained from the authors upon reasonable request.

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
