# Peer review of "Femtosecond Laser Modification of Silica Optical Waveguides for Potential Bragg Gratings Sensing"

_materials, 2022, doi:10.3390/ma15186220_

Round 1

Reviewer 1 Report

In the paper, entitled ‘Femtosecond laser modification of silica optical waveguides for Bragg gratings sensing’, the authors demonstrate the femtosecond laser direct writing of Bragg gratings on silica optical waveguides with flame hydrolysis deposition of SiO2. The results are well presented and the manuscript is well written. However, mechanism of the change of refractive index of SiO2 prepared by FHD is not clear. Therefore, for this paper to be accepted, the authors are supposed to address the following comments properly.

Comments

1 The novelty of this manuscript is to use SiO2 deposited by FHD for making Bragg grating by femtosecond laser writing. However, the mechanism is still unknown, and it is not rigorous to guess the mechanism by ‘may be similar to fused silica’, as shown in Line 208. The authors are supposed to explore and make it clear, to support the novelty.

2 What is the advantages of using FHD deposited SiO2?

3 The authors claim that silica waveguides have low polarization dependent loss. Can the authors also simulate the refractive index of the TM mode? Does the proposed bragg grating also work for the TM polarization, and did the authors measure it to demonstrate low polarization dependent loss ?

4 In line 94, the authors claim that the refractive index of TE mode is almost the same as that of the fiber, resulting in a low coupling loss of 0.2 dB. I think it is not rigorous, as the coupling loss is jointly determined by the mode overlap and the Fresnel reflection.

5 What is the refractive index of SiO2 deposited by flame hydrolysis deposition before and after the femtosecond laser writing? What parameters have been applied to Eq 1 for calculating the periods and the duty cycles?

6 The authors use ASE as the light source to characterize the transmission or reflection. Normally, ASE is kind of noise and unpolarized light. How did the author control the polarization during the experiment?

7 How to improve the bandwidth and the extinction ratio of the bragg grating transmission or reflection spectrum?

8 What is the fabrication resolution with this method?

9 I have a little doubt about the schematic of WDM shown in Fig 6. U shaped channels are placed to couple the reflected light. However, as the input light is incident in the main waveguide, the forward light can also couple into the U shaped channels, and terminate at the forward end. If so, no light will reach the bragg grating. Can the authors provide some simulations to demonstrate the proposed design?

Author Response

We thank Reviewer 2 for his/her enlightening suggestions and recommendations. We have thoroughly addressed all of the comments in the revised manuscript, also as presented in the attached response file.

Reviewer 2 Report

In this work, authors fabricated third order Bragg gratings on silica optical waveguides with the femtosecond laser direct writing technique. The transmission and reflection spectra of the Bragg grating were also measured. Furthermore, the authors fabricated cascaded Bragg gratings and measured the transmission spectra of the Bragg grated waveguides.

1.     Line 71-73: Authors mentioned that “But the post-processing technology of the silica-based PLC waveguide by femtosecond laser direct writing technology has not yet been reported to the best of our knowledge.” But authors did not report any post-processing technology in this given MS. Comment.

2.     Line 75-76: “Compared with the fiber Bragg grating, the waveguide case can be easily prepared without any aberration correction.” What is that exactly being referred here.

3.     Line 86-88: Did authors used commercially available silicon waveguides? Give details. Give the details of silica chip as well.

4.     Line 121: “ending” could be replaced with “blocking of the pulse train”.

5.     The experimental schematic needs to be updated to an accurate setup matching to the experimental details.

6.     Define terms such as duty cycle? Did authors verify the optimized duty cycle to obtain the Bragg structures?

7.     How does the focusing point ensured each and every time? Authors mentioned that the focusing position was above the waveguide.

8.     Line 148-149: What is the spot size obtained with the objective used? What is the working distance of it? How accurate were the focusing conditions?

9.     Line 159: How duty cycle and writing length are related? Are they? Explain.

10.  Line 159-160: “During the grating writing, the transmission spectra can be monitored. And the extinction ratio of the transmission peak increases with the total length of the gratings.” Is that what authors observed or is it an observation reported by other researchers?

11.   Line 169: “fabrication condition such as the pulse width, laser energy, focusing position, etc.” Please cite the literature and describe how these parameters play their role.

12.  Line 177: Keep the both scales common, so that it gives more better understanding.

13.  Can authors show the real-time application of these gratings as described in section 4?

Author Response

(The authors gave the same response as above.)

Round 2

Reviewer 1 Report

The authors answered sincerely to the reviewer's comments, and made necessary modifications to their manuscript. I could recommend this manuscript to be accepted by Materials.

Reviewer 2 Report

Authors have explained all the comments and rephrased a lot of sentences for the better readability and understanding.

The MS can be accepted for the publication, in this revised form